# The Role of Chemokines in the Development of Gastric Cancer—Diagnostic and Therapeutic Implications

**DOI:** 10.3390/ijms21228456

**Published:** 2020-11-10

**Authors:** Elzbieta Pawluczuk, Marta Łukaszewicz-Zając, Barbara Mroczko

**Affiliations:** 1Department of Neurodegeneration Diagnostics, Medical University of Bialystok, 15-269 Bialystok, Poland; elzbieta.pawluczuk16@wp.pl; 2Department of Biochemical Diagnostics, Medical University of Bialystok, 15-269 Bialystok, Poland; marta.lukaszewicz-zajac@umb.edu.pl

**Keywords:** chemokines, tumor markers, gastric cancer, chemokine receptor

## Abstract

Gastric cancer (GC) is the fifth most common cancer worldwide and the second leading cause of cancer-related death. GC is usually diagnosed at an advanced stage due to late presentation of symptoms. Therefore, there is a need for establishing more sensitive and specific markers useful in early detection of the disease when a cancer is asymptomatic to improve the diagnostic and clinical decision-making process. Some researchers suggest that chemokines and their specific receptors play an important role in GC initiation and progression via promotion of angiogenesis, tumor transformation, invasion, survival and metastasis as well as protection from host response and inter-cell communication. Chemokines are small proteins produced by various cells such as endothelial cells, fibroblasts, leukocytes, and epithelial and tumor cells. According to our knowledge, the significance of chemokines and their specific receptors in diagnosing GC and evaluating its progression has not been fully elucidated. The present article offers a review of current knowledge on general characteristics of chemokines, specific receptors and their role in GC pathogenesis as well as their potential usefulness as novel biomarkers for GC.

## 1. Gastric Cancer

According to the World Health Organization (WHO) data published in 2018, gastric cancer (GC) is the fifth most common cancer worldwide and the second leading cause of death among cancers [1]. The most common type of GC is adenocarcinoma—a malignant epithelial tumor with glandular origin. Anatomically, it is classified as cardia and non-cardia adenocarcinoma. Gastric cardia adenocarcinoma shares common characteristics and risk factors with esophageal adenocarcinoma because of the anatomical proximity to the esophagus, while non-cardia cancers are located distally, and their risk factors are more typical of GC [2]. The Lauren classification divides gastric adenocarcinoma into two main histological types: diffuse and intestinal type [3].

Prognosis of GC is poor, as the five year survival rate is lower than 30% [3]. According to a Japanese analysis, incidence of GC is steadily decreasing, with a decline observed particularly among young individuals with the non-cardia type of cancer. On the other hand, an American study reports an increasing incidence of corpus (non-cardia) GC [4]. Numerous risk factors for stomach cancer have been identified [2], including environmental and genetic causes [4]. The main risk factor is *Helicobacter pylori (H. pylori)* infection, which can lead to acute gastritis. *H. pylori* is thought to be responsible for 80% of gastric ulcers. Alcohol consumption and smoking also increase stomach cancer risk. Other factors implicated in GC development are chemical exposure, high temperatures, or working in wood, metal, and rubber processing plants. Additionally, a diet high in salt can break the gastric mucosal barrier and lead to the inflammatory process. Further factors implicated in GC development are obesity—an analysis revealed an increased risk for individuals with a body mass index (BMI) higher than 25 kg/m^2^—and a diet lacking in fruit and vegetables [5]. Gastric surgery is also a predisposing factor. Additionally, exposure to radiation, particularly in the abdominal area, can make individuals more susceptible to GC. Some studies suggest that Epstein–Barr virus may play a role in the development of stomach cancer. Furthermore, individuals with type a blood have a higher risk of developing GC, while those with type 0 blood have a higher risk of developing gastric ulcers. GC is more common in males than in females, which may be associated with the protective influence of hormones such as estrogen or gender differences in dietary habits [2].

Measurement of serum concentrations of biochemical markers such as cancer antigen 72-4 (CA 72-4) and carcinoembryonic antigen (CEA) is a very important diagnostic tool in routine clinical practice. Cancer antigen 72-4 (CA 72-4) is a first-line classical tumor marker with the highest diagnostic sensitivity for GC. Its concentration may also be elevated in patients with colon, pancreatic, breast, or ovarian cancer [6]. A study performed by Joypaul et al. demonstrated that CA 72-4 is a reliable marker in GC and that serial sampling may help to identify recurrence early [7]. The role of CA 72-4 as a screening marker in healthy individuals is not clear. A study by Hu et al. evaluating the utility of CA 72-4 as a screening tool for GC in a healthy population showed that the marker may not be effective in asymptomatic individuals due to low positive predictive values [6]. Another classical tumor marker for GC is CEA, whose levels may be elevated in diseases such as GC, colorectal cancer, pancreatic cancer, lung cancer, or as a result of inflammation. Measurement of serum concentrations of carbohydrate antigen (CA 19-9) is also routinely used in GC diagnosis, although its concentrations may also be elevated in the blood of patients with pancreatic cancer, cholangiocarcinoma, colon cancer, esophageal cancer, hepatocarcinoma, as well as pancreatitis, biliary tract diseases, or liver cirrhosis [8]. In addition, human epidermal growth factor receptor 2 (HER2) is used to classify patients with advanced GC for treatment with trastuzumab. The markers, which are presented above, lack sufficient specificity and are not useful as screening tools aimed at detecting GC at an early stage. Measurement of the levels of these proteins can be used to monitor tumor progression as well as assist in selecting the most appropriate treatment method [9]. Therefore, there is a need for more specific markers in GC diagnosis. Some authors indicate that chemokines may be a group of such biochemical markers, and therefore more intensive research on these molecules is required [8].

### 1.1. Chemokines

Chemokines are proteins secreted in pathological circumstances by various cells such as endothelial cells, fibroblasts, epithelial cells, leukocytes, and tumor cells [10]. They are grouped according to two classifications. In the first classification, there are four classes of chemokines: CC (C-C motif chemokines), XC (X-C motif chemokines), CXC (C-X-C motif chemokines), and CX3C (C-X3-C motif chemokines), where C reflects the position of key cysteine and X represents any amino acid. The most important factor in the second classification is function of the chemokine— inflammatory, homeostatic, or dual-function chemokines are distinguished [11,12].

Chemokines act via their specific receptors. There are over twenty chemokine receptors which are seven-transmembrane, G protein-coupled receptors [13]. One chemokine can be related to many receptors, whereas a specific receptor may be activated by several ligands [14]. Chemokine receptors participate both in physiological processes and in cancers, cardiovascular diseases, or infections [13].

Chemokines are primarily involved in inflammation. Controlled inflammation is an important and beneficial process since it can, for example, promote wound healing or defend the body against pathogens. Moreover, chemokines are also crucial to processes such as atherosclerosis, autoimmune diseases, or HIV infection [14]. They play an important role in autoimmune diseases such as rheumatoid arthritis. These proteins participate in cell recruitment, for example, recruitment of Th1 lymphocytes (T helper cells 1) to the synovium. Blocking receptors involved in this process reduces inflammation. Chemokines secreted in patients with rheumatoid arthritis differ between phases of arthritis. CCL4 (CC chemokine ligand 4), CXCL4 (CXC chemokine ligand 4), CXCL7 (CXC chemokine ligand 7), and CXCL13 (CXC chemokine ligand 13) are expressed in the early stage, while CCL3 (CC chemokine ligand 3) and CCL9 (CC chemokine ligand 9) are expressed in more advanced stages of the disease [15]. On the other hand, uncontrolled inflammation leads to pathological processes and may promote tumor development. Chemokines are associated with cell migration during inflammation and if this process is prolonged, it may lead to carcinogenesis [11].

### 1.2. Chemokines in Cancer

GC is a solid tumor which consists of stromal cells, e.g., endothelial cells or fibroblasts, and is infiltrated by lymphocytes, neutrophils, and macrophages. All of these cells participate in chemokine production [16]. Recent studies have demonstrated that chemokines and their receptors may be involved in tumor initiation and progression. Chemokines coordinate multiple intercellular processes and play a role in tumor progression [14].

Chemokines act indirectly by participating in angiogenesis or directly by affecting tumor transformation, growth, invasion, survival, and metastasis [11]. They are involved in migration, intercellular communication, proliferation, protection from the host response, angiogenesis, and extravasation [13]. Chemokines originate from the tumor microenvironment (TME). Tumor growth is not an independent factor, and this theory has been proposed by a number of researchers [17,18]. Chemokines originating from the TME can facilitate tumor progression or remodeling of TME and signal transduction [19]. Some studies have demonstrated that tumor cells can produce chemokines or express their receptors. Identifying chemokines involved in the progression of various cancers may also be important in future pharmacotherapy of patients. Gao et al. studied CXCL11 expression in colorectal cancer tissue, and they observed that CXCL11 suppression inhibited invasion and cell migration in vitro, while in vivo down-regulation of this chemokine reduced cell growth and metastasis [20]. These proteins may play a crucial role in the pathogenesis of many cancers such as melanoma, breast cancer, ovarian cancer, colorectal cancer, esophageal cancer, as well as gastric cancer [21,22,23].

## 2. Chemokines in Gastric Cancer

### 2.1. CXC Chemokines

Some clinical investigations have indicated that selected chemokines, particularly the CXC family of chemokines, and their specific receptors play an important role in GC pathogenesis (Figure 1). Chen et al. assessed the concentrations of CXCL1, CXCL2, CXCL5, CXCL8, CXCL11, and CXCL13 in tumor drainage blood and peripheral blood from relapse-free GC patients and found that post-treatment levels were lower in comparison to pre-treatment concentrations [16]. Reduced CXCR1 (CXC chemokine receptor 1) and CXCR3 (CXC chemokine receptor 3) expression was associated with a smaller tumor size and lower tumor stage (TNM) stage, while decreased CXCR2 (CXC chemokine receptor 2) and CXCR4 (CXC chemokine receptor 4) expression correlated with a larger tumor and higher TNM stage. Patients with recurrent GC had higher concentrations of CXCL1, CXCL2, CXCL4, CXCL5, CXCL7, CXCL8, CXCL9, CXCL10, CXCL12, CXCL13, and CXCL14 in tumor drainage blood and peripheral blood compared to relapse-free patients. The authors suggest that selected CXC chemokines may be used as markers of GC development and progression [16] (Table 1).

In a study performed by Lim et al., serum CXCL12 and CXCL5 concentrations in four groups of patients—normal-risk, high-risk of GC (intestinal metaplasia or adenoma), early GC and advanced GC—were analyzed. Both CXCL12 and CXCL5 concentrations were elevated in the group with advanced GC and correlated with the presence of distant metastasis. Serum CXCL12 levels were associated with nodal involvement while CXCL5 concentrations were associated with depth of tumor invasion and distant metastasis. Furthermore, the receiver operating characteristic (ROC) curve showed that serum CXCL5 and CXCL12 levels had higher diagnostic significance in predicting GC compared to CEA. The researchers suggested that the combined measurement of CXCL5, CXCL12, and CEA concentrations may be used to predict GC and distant metastasis [24]. Moreover, a study by Park et al. revealed that enhanced CXCL5 expression was correlated with nodal involvement (N-stage), while there was no relationship between CXCL5 immunoreactivity and tumor size or direct extent of the primary tumor. In addition, serum CXCL5 concentrations were higher in patients with advanced GC in comparison to those with benign tumors, which may suggest a potential role of CXCL5 in GC progression [25].

In addition, Yamamoto et al. reported that CXCL1 expression was statistically associated with age, higher T-stage, venous and lymphatic invasion, and metastasis to the lymph nodes, whereas CXCL2 levels were correlated with lower T-stage and no nodal involvement. Moreover, CXCL3 expression was associated with lower infiltration of tumor cells while CXCL7 expression was associated with a more advanced age, presence of lymph node metastasis, and venous invasion. There were no statistically significant differences between CXCL5, CXCL6, and CXCL8 levels and clinicopathological characteristics of GC [26].

CXCL8, also known as IL-8, is a chemokine which promotes inflammation and stimulates vascular growth factor expression (VEGF) and cell proliferation. CXCL8 can be produced by macrophages, monocytes, and endothelial and epithelial cells. In *H. pylori* infection, CXCL8 is produced by gastric epithelial cells. The expression of this chemokine can be measured in gastric antral biopsy specimens from patients with *H. pylori* infection or gastritis. CXCL8 attracts neutrophils to the site of inflammation. However, these neutrophils are unable to eliminate *H. pylori* infection, which leads to a chronic neutrophil inflammation process and gastritis. *H. pylori* and chronic inflammation are factors contributing to GC development [27]. A study by Baj-Krzyworzeka et al. revealed elevated CXCL8 concentrations in GC patients. Thus, this chemokine facilitates tumor growth by neutrophil chemotaxis and promotes formation of metastases via the production of metalloproteases and vascular endothelial growth factor [10]. Haghazali et al. demonstrated that CXCL8 concentration was higher in a group of patients with GC in comparison to the control group and individuals with peptic ulcer disease. Moreover, elevated CXCL8 concentration correlated with *H. pylori* infection [28]. Lee et al. discovered that serum CXCL8 concentration was higher in patients with *H. pylori* infection and GC in comparison to patients with *H. pylori* infection but without GC [29].

### 2.2. CCL Chemokines

It has been indicated that the CCL family of chemokines may play an important role in GC pathogenesis (Table 2). In a study by Baj-Krzyworzeka et al., plasma concentrations of CCL2, CCL3, CCL4, CCL5, CXCL8, CXCL9, and CXCL10 in GC were assessed. There were significant differences between plasma CCL2, CCL4, and CCL5 levels in GC patients and healthy controls. CCL2 and CCL10 concentrations correlated with clinical stage of the disease and were highest in patients with more advanced cancer. However, no associations were found between concentrations of these chemokines and histopathological characteristics of GC [10]. The authors concluded that increased levels of these chemokines promote migration and invasiveness of GC cells [10].

Some clinical investigations have revealed that patients with GC have a higher serum concentration of CCL5 in comparison to healthy controls, which has been confirmed by other authors [30,31,32]. The overall survival rates of GC patients with elevated serum CCL5 levels were reduced in comparison to subjects with low concentrations of this chemokine. Elevated CCL5 concentrations correlated with poorer histological differentiation, greater depth of tumor invasion, more frequent lymph node involvement, and advanced tumor stage [30]. In a study by Wang et al., increased serum concentration of CCL5 correlated with a higher T- and N- stage, peritoneal metastasis and decreased survival. The authors concluded that CCL5 may participate in the promotion, invasion, and peritoneal metastasis of GC and suggested that this chemokine may be a therapeutic target in the future [31]. The findings were confirmed by Ding et al., who also reported that CCL5 may be a new target in the treatment of GC patients [32].

In a study by Tao et al., CCL2 expression was elevated in GC specimens. The overall survival rate of GC patients with elevated CCL2 expression was reduced in comparison to individuals with lower CCL2 expression. According to the authors of the study, CCL2 may be used as an independent prognostic marker for GC [33].

### 2.3. Chemokine/Receptor Axis

An increasing body of evidence suggests that chemokine/specific receptor pathways are involved in the pathogenesis of many malignancies, including GC (Table 3). There are studies demonstrating that CCL20 affects the progression of colorectal cancer, liver cancer, pancreatic cancer, breast cancer, and GC [19,34,35,36]. CCL20, in contrast to other chemokines, binds only CCR6. Chen et al. reported that CCR6 expression in cancer is elevated and connected with malignant progression. The upregulation of CCR6 has been observed in patients with GC in comparison to normal gastric tissue. CCL20 recruits other cells to the TME, which can make the process of signaling more complicated [19]. Jin et al. reported that CXCR6 expression was upregulated in GC tumor tissue and was significantly correlated with lymph node and distant metastases as well as an advanced clinical stage of GC. A larger tumor enhanced CEA concentration and a higher TNM stage enhanced CXCR6 expression were correlated with worse overall survival rates. The authors also performed a three year follow-up of patients with GC which revealed that patients with lower CXCR6 levels had longer overall survival than those with the elevated level of this cytokine [37].

Growing evidence shows the role of the CXCL12 and the CXCR4 pathways in GC pathogenesis. Xu et al. revealed that expression of *CXCL12* and CXCR4 was significantly different between GC tissue and normal gastric tissue. However, no statistically significant differences in CXCL12 and CXCR4 expression were detected between stages of GC [38]. Rubie et al. demonstrated statistically significant downregulation of CCL12 and upregulation of CXCR4 mRNA in GC tissue compared to normal gastric tissue. Furthermore, it was observed that CXCL12 mRNA was related to a more advanced tumor stage. In patients who received neoadjuvant chemotherapy, CXCR4 expression was upregulated, while downregulation in CXCL12 expression was observed in subjects with lymph node and vein involvement [39]. In addition, Lee et al. assessed the role of the CCR4/CCL17 axis in GC development. The authors used clinical samples of GC and GC standard cell lines and established that 75% of GC cell lines and 17% of clinical samples expressed CCR4. Patients with CCR4 positive tumors were found to have a worse prognosis in comparison to those with CCR4 negative tumors. The authors concluded that the CCR4/CCL17 axis is associated with recurrence of GC and decreased overall survival of patients with this malignancy [40].

### 2.4. Chemokine Receptors

It has been proven that chemokine receptors may also play an important role in GC pathogenesis, and therefore this issue is frequently examined in clinical studies (Table 4). In a study by Chen et al., the expression of chemokine receptor 3 (CXCR3) was significantly enhanced in GC tissue in comparison to paracancerous tissue and was related to higher differentiation and smaller depth of invasion of GC. The authors suggested that elevated CXCR3 expression could be used as a prognostic marker for GC [41].

Li et al. established that overexpression of CXCR1 and CXCR2 was associated with the presence of distant metastases, tumor differentiation and advanced stage of [42]. These results suggest that CXCR1 and CXCR2 play a crucial role in GC progression [42]. Moreover, in a study by Yamamoto et al., the expression of CXCR2 ligands such as CXCL1, CXCL2, CXCL3, CXCL5, CXCL6, CXCL7 and CXCL8 in GC tissue was determined [26]. CXCL1 expression was correlated with greater depth of invasion, whereas lymph node metastasis was correlated with increased CXCL1 and CXCL7 expression [26]. Prognosis of CXCL1-positive patients was significantly poorer in comparison to that of CXCL1-negative patients. The authors concluded that among CXCR2 ligands, CXCL7 and CXCL1 may play an important role in GC progression via CXCR2 signaling [26].

In a study by Nambara et al., expression of CXCR7 in GC was higher than in normal gastric tissue [43]. The overall survival was decreased in patients with elevated CXCR7 mRNA expression. Enhanced CXCR7 expression was also associated with lymph node metastasis, venous invasion, advanced TNM stage, deeper tumor invasion, and poor differentiation of GC. There were no statistically significant differences between CXCR7 expression and patients’ gender, age, and lymphatic invasion. Moreover, higher CXCR7 expression was correlated with angiogenesis as well as with migration and proliferation of GC cells [43].

Some clinical investigations have indicated the significance of the aberrant expression of chemokine receptors in GC, including copy number changes or upregulation of chemokine receptors mRNA. In recent years, several cancer parameters, such as immune infiltration, somatic copy number alterations (SCNA), tumor mutation burden (TMB), tumor purity, cytolytic activity (CYT), and drug sensitivity have been reported to be prognostic factors and potential therapeutic markers for various cancers, including GC [44,45]. There was a statistically significant correlation between CXCR4 levels and TMB, CYT, tumor purity, tumor immune infiltration, and drug sensitivity. Moreover, elevated CXCR4 expression can affect the resistance of cancer cells to drugs [44]. What is more, a possible link between tumor immunity and aberrant expression of CCR4 in GC cells has been proved [46]. The study of Yang et al. has revealed that the aberrant expression of CCR4 in GC could contribute to tumor-induced immunosuppression [46]. It has been proved that TNF alpha (tumor necrosis factor alpha) and transcription factor NF-jB are the key factors that connect inflammation to tumor promotion [46,47,48]. A significant positive relationship between expression of CCR4 and TNF alpha in GC tissue has been found. Thus, there is a possibility that TNF-alpha might upmodulate expression of CCR4 in an NF-jB-dependent manner. Authors conclude that CCR4 might play a novel function in immunosuppression by regulating secretion of cytokine and cytotoxicity of immune cells. Moreover, TNF alpha promotes tumor-induced immunosuppression via induction of aberrant CCR4 expression and may represent a new target for immune treatment of GC in the future [46]. The study of Cheng et al [49] revealed that CXCL12/CXCR4 induces GC cell epithelial–mesenchymal transition (EMT), what is associated with activation of *MET proto-oncogene (c-MET)*. CXCR4 expression is positively correlated with *c-MET* phosphorylation, and the presence of both correlates with poor GC prognosis. Authors indicate the importance of CXCR4 and *c-MET* in GC metastasis and suggest that targeting specific molecular components of their signaling pathways will provide new opportunities for GC treatment [49]. The study of Xing et al [50] demonstrated that CXCL16 mRNA expression was elevated in cancer tissue compared with adjacent mucosa, whereas CXCR6 was expressed in the opposite manner. In addition, expression of CXCL16 inversely correlated with tumor stage, the invasion depth of the tumor and lymphatic invasion, what may suggest that this chemokine and its receptor CXCR6 may play a role in gastric tumorigenesis [50].

### 2.5. The Importance of Interaction of Chemokines and Their Specific Receptors in GC Progression

Chemokines are able to bind the extracellular domain of chemokine receptors, which consists of the N-terminus and extracellular loops. During the activation, the intracellular domains dissociate from G-proteins, composed of three subunits such as α, β, and γ heterotrimers [11]. The result of this process is the formation of the second messengers inositol triphosphate (IP3) and diacylglycerol (DAG), leading to the activation of multiple downstream signaling cascades and cytoplasmic calcium mobilization, including phosphatidylinositol 3-kinase (PI3K)/Akt, Ras/mitogen-activated protein kinase (MAPK), and Janus kinase/signal transducer and activator of transcription (JAK/STAT) pathways [11,51,52]. Abnormal activation of PI3K/Akt and Wnt/β-catenin pathways has been found to promote GC progression and poor prognosis of cancer patients, suggesting their potential therapeutic target for GC patients [11,53,54,55]. It has been shown that MAPK pathways alterations may lead to GC, including *H. pylori* MAPK-triggering to transform from gastric normal epithelium to GC development. Moreover, the impact of the MAPKs in GC progression and metastasogenesis and the involvement of the dysregulated kinase pathways in GC encourage new studies on therapeutic drugs which might improve the survival of GC patients [55]. The study of Ji et al. [56] demonstrated an immunoregulatory role of β-catenin signaling in GC and suggested the therapeutic potential of CCL28 blockade for the treatment of GC, because this chemokine was proved to be as a key linker between the oncogenic β-catenin signaling and the GC microenvironment. 

Based on presented investigations, chemokines and their specific receptors may be used as targets for biopharmaceuticals and improve treatment process of patients with many diseases, including GC. The CXCR4/CXCL12 axis was taken into consideration as a potential factor in GC therapy [39]. One of the chemokine-targeting drugs already used in other diseases is Plerixafor [57]. This substance blocks CXCR4 from binding to CXCL12. It enables mobilization of hematopoietic stem cells to peripheral blood and in result can be used for transplantation of hematopoietic stem cells in diseases such as lymphomas and multiple myeloma. [57]. Moreover, serum levels of chemokines may become predicting factors for response to treatment or resistance to drugs. The results of Zhai et al. [58] presented that higher serum level of CXCL8 mediated resistance to platinum-based antineoplastic drugs in GC patients [58]. CXCR2 promotes the process of metastasogenesis and tumor growth, while the two chemokines that bind to CXCR2 (CXCL1 and CXCL5) are released by macrophages, and this process is supported by GC cells. Thus, this receptor can also be a good target in developing new strategies of GC treatment [59].

## 3. Conclusions

GC is the fifth most common cancer in the world. Therefore, non-invasive methods of diagnosing GC at an early stage are necessary. Selected chemokines and their specific receptors play a potential role as future biomarkers which can be used in the process of diagnosing GC and selecting appropriate treatment for patients. There is a need for more research on a larger cohort of GC patients to determine which chemokines are the most specific for GC and which of them might be used as potential tumor biomarkers for GC in the future.

## Figures and Tables

**Figure 1 ijms-21-08456-f001:**
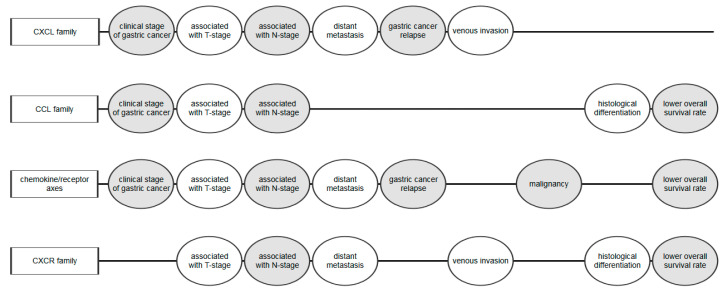
Correlations between the role of chemokines and features assessed in gastric cancer (GC) [10,16,19,24,25,26,27,28,29,30,31,32,33,34,35,36,37,38,39,40,41,42,43]. Rectangles represent chemokine ligand (CXCL and CCL) families, chemokine/receptor pairs, and chemokine receptors (CXCR) families analyzed in the study. Ellipses represent selected features in gastric cancer correlated with concentration or expression of chemokine families, chemokine/receptor pairs, and chemokine receptors. T-stage—depth of tumor invasion. N-stage—presence of lymph node metastasis.

**Table 1 ijms-21-08456-t001:** The role of CXCL (CXC chemokine ligand) in gastric cancer (GC).

Chemokines	Source	Results	References
**CXCL1**	Concentration in tumor drainage blood and peripheral blood	Lower concentration after treatment Higher concentration in GC relapse	[16]
	Expression	Associated with higher T-stage, venous and lymphatic invasion, age, and metastasis of lymph nodes	[26]
**CXCL2**	Concentration in tumor drainage blood and peripheral blood	Lower concentration after treatment Higher concentration in GC relapse	[16]
	Expression	Associated with lower T-stage	[26]
**CXCL4**	Concentration in tumor drainage blood and peripheral blood	Lower concentration after treatment Higher concentration in GC relapse	[16]
**CXCL5**	Concentration in tumor drainage blood and peripheral blood	Lower concentration after treatment Higher concentration in GC relapse	[16]
	Serum concentration	Elevated in advanced GC, correlated with presence of distant metastasis and T-stage	[24]
		Higher in IIIB and IV stages of GC than in benign conditions	[25]
	Expression	Correlated with N-stage, higher in N2 and N3	[25]
**CXCL7**	Concentration in tumor drainage blood and peripheral blood	Higher concentration in GC relapse	[16]
	Expression	Associated with older age, presence of metastasis, and invasion of the lymph nodes, venous invasion and negative cytology of peritoneum	[26]
**CXCL8**	Concentration in tumor drainage blood and peripheral blood	Lower concentration after treatment Higher concentration in GC relapse	[16]
	Serum concentration	Elevated in advanced stage of GC	[10]
		Higher in patients with *H. pylori* and GC	[27]
		Higher in GC than peptic ulcer disease and control group	[28]
		Higher concentration correlated with *H. pylori* infection	[28,29]
**CXCL9**	Concentration in tumor drainage blood and peripheral blood	Lower concentration after treatment Higher concentration in GC relapse	[16]
**CXCL10**	Concentration in tumor drainage blood and peripheral blood	Lower concentration after treatment Higher concentration in GC relapse	[16]
**CXCL11**	Concentration in tumor drainage blood and peripheral blood	Lower concentration after treatment	[16]
**CXCL12**	Concentration in tumor drainage blood and peripheral blood	Higher concentration in GC relapse	[16]
	Serum concentration	Elevated in advanced GC, correlated to presence of distant metastasis and nodal involvement	[24]
**CXCL13**	Concentration in tumor drainage blood and peripheral blood	Lower concentration after treatment Higher concentration in GC relapse	[16]
**CXCL14**	Concentration in tumor drainage blood and peripheral blood	Higher concentration in GC relapse	[16]

*H. pylori*—Helicobacter pylori. T-stage—depth of tumor invasion. N-stage—presence of lymph node metastasis.

**Table 2 ijms-21-08456-t002:** The role of CCL chemokines in gastric cancer (GC)

Chemokines/Receptors of Chemokines	Source	Results	References
CCL2	Expression	Elevated in 66% of GC specimen, correlated with lower overall survival rate	[33]
	Plasma concentration	Correlated with clinical stage of GC	[10]
CCL5	Serum concentration	Higher in GC than control group, overall survival reduced when elevated, elevated concentration correlated with more advanced stage of the tumor, higher depth invasion, low histological differentiation and lymph node involvement	[30]
		Increased concentration correlated with higher T-stage, N-stage, peritoneal metastasis and decreased survival	[31]
CCL10	Plasma concentration	Correlated with clinical stage of GC	[10]

T-stage—depth of tumor invasion. N-stage—presence of lymph node metastasis.

**Table 3 ijms-21-08456-t003:** The role of chemokine/receptor axis in gastric cancer (GC).

Chemokines/Receptors of Chemokines	Source	Results	References
CCL20/CCR6	Expression	Elevated in GC and related to malignancy	[19]
		Upregulated in GC tumor tissues, correlated with lymph node and distant metastases, advanced clinical stage of GC, larger tumor, worse overall survival	[37]
CXCL12/CCR4	Expression	Higher in GC tissues	[38]
		Correlated with more advanced tumor stage, upregulated after neoadjuvant chemotherapy	[39]
CCL17/CCR4	Expression	Elevated in GC tumor cells, indicates worse prognosis, associated with relapse of GC and decreased overall survival	[40]

**Table 4 ijms-21-08456-t004:** The role of chemokine receptors in gastric cancer (GC).

Chemokines/Receptors of Chemokines	Source	Results	References
CXCR1	Expression	Associated with presence of distant metastasis, tumor differentiation and advanced stage of GC	[42]
		Reduced expression associated with smaller tumor and lower TNM stage	[16]
CXCR2	Expression	Associated with presence of distant metastasis, tumor differentiation and advanced stage of GC	[42]
		Reduced expression correlated with large tumor and higher TNM stage	[16]
CXCR3	Expression	Higher in GC tissue than paracancerous tissue, related to higher differentiation, smaller depth invasion, longer overall survival and lower mortality rate	[41]
		Reduced expression associated with smaller tumor and lower TNM stage	[16]
CXCR4	Expression	Reduced expression correlated with larger tumor and higher TNM stage	[16]
CXCR7	Expression	Higher in GC, overall survival was lower, connected with lymph node metastasis, venous invasion, advanced TNM stage, deeper invasion of the tumor and poor histological differentiation	[43]

GC—gastric cancer. TNM—tumor stage (Tumor, Lymph nodes, Metastasis) classification of malignant tumors.

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
