# Peer review of "The Role of Chemokines in the Development of Gastric Cancer—Diagnostic and Therapeutic Implications"

_ijms, 2020, doi:10.3390/ijms21228456_

Round 1
Reviewer 1 Report
In this review manuscript, the authors summarized the chemokines and their specific receptors in gastric cancer initiation and progression. The chemokines are produced by kinds of cells including cancer cells, endothelial cells, fibroblasts and leukocytes, which can serve as diagnostic or even prognostic biomarkers. This review is comprehensive as not many reports described the chemokines in GC. On the contrary, the chemokine receptors (GPCR members) have been largely reported. Several concerns need to be addressed to enrich the content:
1, Please give short abbreviations of CC, XC, CXC and CX3C;
2, The authors highlighted the CXCL and CCL family in GC development and progression. However, the chemokine receptors (G-protein-coupled receptors) are also very important in driving GC and the authors need to briefly induce the aberrant activation of chemokine receptors in GC, such as copy number changes, mutation apart from mRNA upregulation.
3. The interaction of chemokines and receptors activates the downstream signaling, such as MAPK, Akt or beta-catenin pathways, thus to drive gastric carcinogenesis. It will be more comprehensive if the review includes the downstream activation.
Author Response
Department of Neurodegeneration Diagnostics, Medical University
Waszyngtona 15 a, 15-269 Białystok, Poland
phone 48 85 7468587, fax 48 85 7468585, e-mail: zdchn@umb.edu.pl
Białystok, 6 November 2020
Editor-in-Chief
International Journal of Molecular Science
Prof. Dr. Kurt A. Jellinger
Dear Editor,
Thank you very much for having our revised manuscript entitled: “The role of chemokines in the development of gastric cancer – diagnostic and therapeutic implications”. The changes in the revised version of our manuscript have been highlighted (underlined). We hope you will find our answers satisfying and the revised manuscript will be acceptable for the publication.
In case of further questions, do not hesitate to contact me.
Sincerely,
Professor Barbara Mroczko
DETAILED ANSWERS TO THE REVIEWERS’ COMMENTS
Reviewer 1
In this review manuscript, the authors summarized the chemokines and their specific receptors in gastric cancer initiation and progression. The chemokines are produced by kinds of cells including cancer cells, endothelial cells, fibroblasts and leukocytes, which can serve as diagnostic or even prognostic biomarkers. This review is comprehensive as not many reports described the chemokines in GC. On the contrary, the chemokine receptors (GPCR members) have been largely reported. Several concerns need to be addressed to enrich the content:
Please give short abbreviations of CC, XC, CXC and CX3C.- The authors highlighted the CXCL and CCL family in GC development and progression. However, the chemokine receptors (G-protein-coupled receptors) are also very important in driving GC and the authors need to briefly induce the aberrant activation of chemokine receptors in GC, such as copy number changes, mutation apart from mRNA upregulation.
- The interaction of chemokines and receptors activates the downstream signaling, such as MAPK, Akt or beta-catenin pathways, thus to drive gastric carcinogenesis. It will be more comprehensive if the review includes the downstream activation.
Authors' Responses to Reviewer's Comments (Reviewer 1)
Thank you very much for positive review.
- Please give short abbreviations of CC, XC, CXC and CX3C.
Authors' Responses to Reviewer's Comments (Reviewer 1)
Abbreviations of CC, XC, CXC and CX3C have been added in the revised version of the manuscript, according to the Reviewer’s suggestion (page 2, lines 75-76).
- The authors highlighted the CXCL and CCL family in GC development and progression. However, the chemokine receptors (G-protein-coupled receptors) are also very important in driving GC and the authors need to briefly induce the aberrant activation of chemokine receptors in GC, such as copy number changes, mutation apart from mRNA upregulation.
Authors' Responses to Reviewer's Comments (Reviewer 1)
Thank you very much for the interesting suggestion. New paragraph concerning the aberrant activation of chemokine receptors in GC has been added in the “Chemokines in gastric cancer” section in the revised paper, as it was recommended (page 8-9, lines 246-273).
- The interaction of chemokines and receptors activates the downstream signaling, such as MAPK, Akt or beta-catenin pathways, thus to drive gastric carcinogenesis. It will be more comprehensive if the review includes the downstream activation.
Authors' Responses to Reviewer's Comments (Reviewer 1)
Thank you very much for the interesting suggestion. New paragraph concerning the interaction of chemokines and specific receptors that activates the downstream signaling has been added in the “Chemokines in gastric cancer” section in the revised version of the paper, according to Reviewer’s 1 suggestion (page 9, lines 246-304).
Reviewer 2 Report
In this review, Pawluczuk et al. have discussed the therapeutic and diagnostic implications of chemokines for the gastric cancer.
The authors have covered the background in gastric cancer and chemokines both in the review. Then the authors have discussed the specific family of chemokines and how they regulate the Gastric cancer development and if it has any role in the relapse. This provides great overview of all the findings available.
However, in my opinion, this review does not discuss the "Therapeutic" part in great detail. If any drugs that specifically target either CCR directly or the pathway axis should be discussed. That will greatly add the value to this review.
Author Response
Department of Neurodegeneration Diagnostics, Medical University
Waszyngtona 15 a, 15-269 Białystok, Poland
phone 48 85 7468587, fax 48 85 7468585, e-mail: zdchn@umb.edu.pl
Białystok, 6 November 2020
Editor-in-Chief
International Journal of Molecular Science
Prof. Dr. Kurt A. Jellinger
Dear Editor,
Thank you very much for having our revised manuscript entitled: “The role of chemokines in the development of gastric cancer – diagnostic and therapeutic implications”. The changes in the revised version of our manuscript have been highlighted (underlined). We hope you will find our answers satisfying and the revised manuscript will be acceptable for the publication.
In case of further questions, do not hesitate to contact me.
Sincerely,
Professor Barbara Mroczko
DETAILED ANSWERS TO THE REVIEWERS’ COMMENTS
Reviewer 2
In this review, Pawluczuk et al. have discussed the therapeutic and diagnostic implications of chemokines for the gastric cancer.
The authors have covered the background in gastric cancer and chemokines both in the review. Then the authors have discussed the specific family of chemokines and how they regulate the Gastric cancer development and if it has any role in the relapse. This provides great overview of all the findings available.
However, in my opinion, this review does not discuss the "Therapeutic" part in great detail. If any drugs that specifically target either CCR directly or the pathway axis should be discussed. That will greatly add the value to this review.
Authors' Responses to Reviewer's Comments (Reviewer 2)
Thank you very much for positive review.